# Desiccation Tolerance in Moss and Liverwort: Insights into the Evolutionary Mechanisms of Terrestrialization

**DOI:** 10.3390/ijms27010478

**Published:** 2026-01-02

**Authors:** Totan Kumar Ghosh, Anika Nazran, Imran Khan, Shah Mohammad Naimul Islam, Tofazzal Islam, Yuan Xu, Mohammad Golam Mostofa

**Affiliations:** 1Department of Crop Botany, Gazipur Agricultural University, Gazipur 1706, Bangladesh; totan@gau.edu.bd (T.K.G.); anika@gau.edu.bd (A.N.); 2Department of Chemistry, State University of New York College of Environmental Science and Forestry, Syracuse, NY 13210, USA; ikhan@esf.edu; 3Institute of Biotechnology and Genetic Engineering, Gazipur Agricultural University, Gazipur 1706, Bangladesh; naimul@gau.edu.bd (S.M.N.I.); tofazzalislam@gau.edu.bd (T.I.); 4MSU-DOE Plant Research Laboratory, Michigan State University, East Lansing, MI 48824, USA

**Keywords:** bryophytes, desiccation tolerance, adaptation, evolution, growth regulators

## Abstract

As a monophyletic group, bryophytes—mosses, liverworts, and hornworts—represent some of the earliest land plants, evolving under harsh terrestrial conditions that prompted major morphological, physiological, and molecular changes. Limited water availability, extreme temperatures, and osmotic stresses often caused cellular desiccation in these pioneering plants. Because bryophytes occupy a key position in land-plant evolution and are closely related to streptophyte algae, their desiccation-tolerance strategies hold significant evolutionary importance. Early adaptations included changes in growth patterns and the formation of specialized vegetative structures. Bryophytes also survive extreme habitats by regulating physiological and biochemical traits such as photosynthetic pigment maintenance, osmotic adjustment, membrane stability, redox balance, and the accumulation of compatible solutes and stress-responsive proteins. Advances in molecular biology and whole-genome sequencing of model mosses and liverworts have further revealed that they possess diverse stress-responsive signaling components, including phytohormones, receptor proteins, protein kinases, and key transcription factors that control stress-related gene expression. However, a comprehensive synthesis of these molecular mechanisms is still lacking. This review aims to provide an updated overview of how mosses and liverworts use plant growth regulators, stress-responsive proteins, compatible solutes, antioxidants, and integrated signaling networks to survive in dry terrestrial environments.

## 1. Introduction

Rapid shifts in global climate pose an increasing threat to agricultural productivity, with drought emerging as one of the most damaging abiotic stresses faced by terrestrial plants [1,2]. Severe drought leads to cellular desiccation—the near-complete loss of free water—often triggered by combined effects of water limitation, osmotic stress, and freezing temperatures. Understanding how plants cope with desiccation has therefore become a major focus in plant evolutionary and stress biology [3,4,5,6].

Reconstruction of land-plant phylogeny confirms that streptophyte algae are the closest relatives of terrestrial plants. However, the precise placement of bryophytes—mosses, liverworts, and hornworts—remains debated. Recent evidence suggests either that bryophytes form a monophyletic group sister to all other land plants, or that mosses and liverworts form a clade (Setaphyta) with hornworts branching separately [7]. Regardless of this unresolved position, bryophytes occupy a crucial evolutionary stage in early terrestrialization (Figure 1). Their ancestors endured extreme environments, including intense vegetative desiccation reaching cellular water potentials near −100 MPa, equivalent to equilibrium at ~50% relative humidity at 20 °C [8].

Bryophytes, the second-most diverse group of land plants [9], possess simple body plans and a dominant haploid life cycle, making them excellent models for biochemical, physiological, and molecular studies of stress adaptation [10,11]. Most species can survive short periods at cellular water potentials between −20 and −40 MPa, whereas values below −4 to −5 MPa are lethal to most vascular plants [8,12]. Although vegetative desiccation tolerance (DT) is not exclusive to bryophytes, it is more widespread in this group than in tracheophytes, where it has largely been retained only in resurrection plants and reproductive structures of angiosperms [8,13,14] (Figure 1) The emergence of rhizoids—an important evolutionary innovation—further enhanced the ability of early bryophytes to survive harsh terrestrial habitats [15,16].

**Figure 1 ijms-27-00478-f001:**
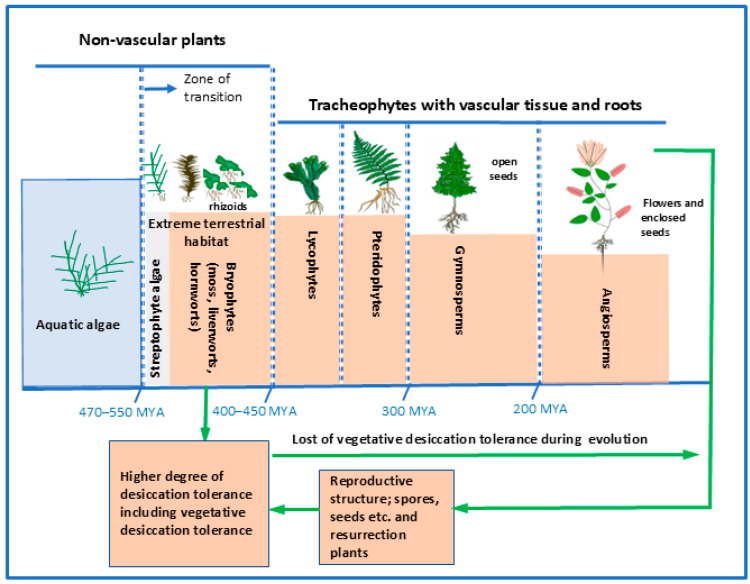
Adaptation and evolution of land plants in extreme terrestrial habitats. During the transition from aquatic to terrestrial environments approximately 470–550 million years ago (MYA), the common ancestors of land plants—streptophyte algae—and the earliest bryophytes encountered extreme conditions, including fluctuating temperatures, water scarcity, osmotic stress, and other environmental challenges. The evolution of rhizoids represented a major innovation, providing anchorage and improving survival in these harsh habitats. Early non-vascular bryophytes developed a high degree of vegetative desiccation tolerance; a trait largely lost in tracheophytes as they evolved true root systems. Nevertheless, the reappearance of vegetative desiccation tolerance in tracheophyte reproductive structures such as spores and seeds, as well as in certain resurrection plants, highlights the conservation of key adaptive mechanisms essential for terrestrial evolution. The timing of these lineage transitions is discussed by Davies et al. [17].

Because of these unique adaptive traits, mosses and liverworts have become key experimental systems for exploring plant adaptation, evolution, and development [3,4,18,19,20]. Their DT strategies involve both morphological and physiological adjustments that protect cells from dehydration, as well as inducible molecular responses mediated primarily by the phytohormone abscisic acid (ABA) and potentially other growth regulators [4,20,21,22].

Although several studies and reviews have addressed aspects of bryophyte DT [21,23], a comprehensive and critical synthesis remains lacking. A deeper understanding of DT in mosses and liverworts is essential for elucidating how early land plants adapted to desiccation-prone environments. Here, we review and critically examine current knowledge of the morphological, physiological, and molecular mechanisms underlying DT in these key bryophyte lineages.

## 2. Desiccation Responses in Mosses and Liverworts

Bryophytes exhibit a wide range of strategies to cope with dehydration, broadly classified into desiccation avoidance and DT mechanisms [22,24,25]. Desiccation avoidance refers to the ability to limit cellular water loss by restricting growth, retaining moisture in vegetative tissues, or entering a dormant state during periods of low water availability [26]. These avoidance strategies often rely on growth suppression or modification of growth-responsive genes under stress [27]. For example, *Bryum bicolorforms* rhizoidal and stem tubers that allow dormancy [24], while *Funaria hygrometrica* produces corm-like bases and bulbils to withstand drought [24]. In *Physcomitrium patens*, exogenous ABA induces the formation of brachycytes—specialized vegetative diaspores that function as desiccation-resistant units [25]. Other species, such as *Sanionia uncinata*, exhibit reduced phyllid and cauloid size during dehydration [28], and *Marchantia polymorpha* shows suppressed growth under ABA and osmotic stress [29].

In contrast, DT is defined as the ability of plants to survive a reduction in cellular water content of more than 90% without sustaining irreversible damage [23]. Many moss tissues can act as long-lived propagules, remaining viable for years under desiccated conditions [24]. Some thalloid liverworts and mosses can endure continuous desiccation for over 20 years [30]. Several bryophyte species exhibit strong constitutive DT. For instance, the gemmae of *M. polymorpha* can survive months of desiccation at room temperature [31], and *P. patens* gametophores tolerate water potentials as low as −100 MPa. Upon ABA treatment, both protonemata and gametophores of *P. patens* show complete DT, equilibrating to approximately −100 MPa [32].

Together, these avoidance and tolerance mechanisms enable mosses and liverworts to persist in extremely harsh environments. This remarkable resilience raises important questions about the underlying regulatory systems. Evidence accumulated over the past decades indicates that numerous proteins and transcription factors (TFs) contribute to DT in bryophytes (Table 1). The roles of these regulatory components are discussed in detail in the following section.

## 3. Roles of Phytohormones in DT and Terrestrialization of Bryophytes

### 3.1. Mosses and Liverworts Rely Heavily on ABA During Their Transition to Terrestrial Environments

ABA, a central regulator of plant physiological processes, is widely recognized as a key stress-responsive phytohormone [54,55]. Under drought, salinity, or osmotic stress, plants typically increase endogenous ABA production, which then activates a suite of adaptive responses [56]. While ABA-mediated drought signaling is well established in angiosperms such as *Arabidopsis thaliana* [57,58], much less is known about how this pathway operates in bryophytes.

In vascular plants, ABA—whether stress-induced or applied externally—initiates a conserved signaling cascade. ABA binds to PYR/PYL/RCAR receptors, promoting their interaction with PP2C phosphatases. In the absence of ABA, PP2Cs deactivate SnRK2 kinases; however, ABA binding inhibits PP2Cs, allowing SnRK2 activation. Activated SnRK2s phosphorylate downstream TFs (AREBs/ABFs), which bind ABRE cis-elements to stimulate stress-responsive gene expression. Numerous TF families, including bZIP, MYB, MYC, NAC, WRKY, and DREB, act downstream of this pathway [6], and ABA signaling also influences membrane transporters, proton pumps, and ion channels [59].

Although bryophyte ABA signaling is less understood, *P. patens* has emerged as a powerful model for evolutionary studies [19,32]. ABA enhances tolerance to freezing, osmotic stress, and desiccation in *P. patens* [22,60], and ABA pretreatment can even induce complete DT in protonemal tissues [61]. Core ABA signaling components—biosynthetic enzymes (PpABA1), receptor (PYL), PP2Cs (ABI1/ABI2), SnRK2, TFs (ABI3, ABI4, ABI5), ABRE elements, and dehydration-related LEA proteins and dehydrins—are all present in this moss (Table 1, Figure 2).

The liverwort *M. polymorpha* has also become a valuable model due to its simple morphology, low-redundancy genome, and amenability to CRISPR/Cas9 editing [3,5]. Its asexual gemmae exhibit remarkable DT [62]. Although ABA signaling is less characterized in liverworts, functional studies of MpABI1, MpPYL, ABREs, and ABA-induced transcripts (*LEA*, *dehydrins*) confirm the presence of an ABA-dependent stress pathway (Table 1). Moreover, ABA activates the MAPKKK Raf (group B3) cascade in both moss and liverwort, enhancing osmotic and desiccation responses via SnRK2 activation [5,40,46]. Notably, *M. polymorpha* possesses multiple B3 and B2 Raf kinases with diverse functions—a contrast to *P. patens*—indicating significant evolutionary diversification of this module. Functional characterization of B2 Rafs in liverworts may offer insight into how land plants adapted to desiccating conditions.

Despite these advances, major gaps remain. Key kinases such as MAPKK and MAPK, as well as essential ABA TFs like bZIPs and SnRK2s, are still poorly defined in *M. polymorpha*. Additionally, ABA was recently shown to regulate sugar-responsive genes to promote vegetative DT [37], underscoring the need to explore sugar-mediated stress pathways in bryophytes, especially given the lack of extensive mutant resources.

DREB also contribute significantly to bryophyte desiccation responses (Figure 2). In *P. patens*, PpDBF1 expression is strongly induced by drought, salt, cold, and ABA [63], indicating that DREB functions in both ABA-dependent and independent pathways in bryophytes (Figure 2). Proteomic analyses further show that ABA enhances the accumulation of defense-related proteins, including RLKs and WRKY TF [64], although their mechanistic roles remain unresolved in bryophytes (Figure 2).

Like angiosperms, bryophytes also exhibit ABA-induced growth arrest [65], but the mechanisms underpinning this avoidance strategy have yet to be fully clarified. ABI3 in *P. patens* (PpABI3) was found to be the negative regulator of the vegetative development possibly by modulating auxin-related genes [66]. In addition, mutation in Raf-like protein kinase in *P. patens* (PpARK) showed no inhibition of growth under exogenous ABA reflecting the role of this kinase in growth arrest [40]. Apart from this, ABA-mediated regulation of expansin in *P. patens* [64] suggests an integrated mechanistic approach for the regulation of growth in early land plants. ABA also supports photoprotection: desiccated tissues of *Pseudocrossidium replicatum* recover photosystem II efficiency (Fv/Fm) more effectively when ABA-treated [67], indicating an ancient role for ABA in safeguarding photosynthetic machinery (Figure 2).

Overall, ABA orchestrates a wide array of signaling components—including receptors, kinases, TFs, and promoter elements—to enhance the expression of stress tolerance genes and promote cellular protection during desiccation of bryophytes (Figure 2). At the same time, ABA limits growth as part of a broader stress avoidance strategy (Figure 2). Future transcriptomic and proteomic studies, particularly in fully desiccation-tolerant bryophytes such as *Syntrichia ruralis*, will be essential to fully resolve the conserved and lineage-specific roles of ABA in bryophyte DT. Hopefully, sequencing of whole genome of *S. ruralis* explored the presence of conserved and novel regulators of DT in plants [68]. The occurrence of MYB TF at the upstream of the ABI3 and LEA proteins in desiccation tolerant moss *S. ruralis* reflect the opportunities for fine-tuning ABA signaling in bryophytes [68].

### 3.2. Role of Ethylene (ET) in the Terrestrial Adaptation of Mosses and Liverworts

Ethylene (ET) functions alongside other phytohormones to regulate plant responses to environmental stress [69]. In seed plants, aminocyclopropane-1-carboxylate (ACC) is the direct precursor of ET, but in non-seed plants ACC also performs ET-independent roles [39]. Although ET- and ACC-biosynthetic pathways remain poorly understood in bryophytes [21], core signaling components—including ET receptors, CTR1, ETHYLENE INSENSITIVE (EIN), and ERF—are conserved in both mosses and liverworts [3,19]. Notably, bryophytes possess ACC synthase (ACS) but lack ACC oxidase (ACO), suggesting that ET signaling may occur largely independent of canonical ET biosynthesis [19].

Evidence of ET-mediated abiotic stress signaling is limited in bryophytes. In *P. patens*, CTR1 negatively regulates ET signaling, functioning similarly to its Arabidopsis counterpart [41]. CTR1 also responds to both ET and ABA, indicating potential crosstalk between these pathways during adaptation to extreme terrestrial environments. Recent work in *M. polymorpha* confirms ET involvement in stress signaling through analyses of MpCTR1 and MpEIN3 [70]. Elevated gemma growth in *Mpctr1* mutants suggests a role for CTR1 in suppressing growth—an important desiccation avoidance strategy [39].

Intriguingly, ACC and ET act antagonistically in bryophytes: ACC inhibits growth or induces dormancy, whereas ET promotes cell expansion, as observed in the aquatic liverwort *Riella helicophylla*. This contrasts with higher plants, where ACC and ET actions overlap. Such divergence implies that ACC may have played an adaptive role in regulating growth and survival in early land plants (Figure 3).

Whether ET or ACC regulate desiccation-responsive transcripts remains unknown. Functional analyses of ET- and ACC-related mutants under desiccation stress will help clarify their ancestral roles and evolutionary interactions with ABA signaling. While ABA–ET crosstalk in drought and osmotic stress is well described in angiosperms [71], similar molecular relationships in bryophytes remain unexplored (Figure 3). Very recent investigation demonstrated that disruption of EIN2 in *P. patens* (*ppein2ab* mutant) showed reduced sensitivity to ABA [72]. The mutant also exhibited higher sensitivity to hyperosmosis, freezing stress and reduced levels of LEA proteins [72] suggesting the positive crosstalk between ABA and ET while addressing DT in bryophytes (Figure 3).

### 3.3. Jasmonic Acid (JA) and Its Role in Moss and Liverwort Terrestrialization

Jasmonates are key regulators of plant growth, development, and stress responses [73]. Liverworts possess orthologs of major JA-signaling components—including COI1, jasmonate ZIM-domain (JAZ), novel interactor of JAZ (NINJA), and MYC—indicating that jasmonate responses originated early in land plant evolution [3,74]. However, bryophytes lack the canonical jasmonoyl-L-isoleucine (JA-Ile) biosynthetic machinery, raising questions about how the pathway operates in these lineages.

A significant breakthrough was the identification of dn-OPDA, a JA-Ile precursor, as a bioactive ligand for MpCOI1 in *M. polymorpha* [74]. Transcriptomic analyses of *Mpmyc* mutants further revealed the presence of key OPDA biosynthetic genes (*MpAOC*, *MpAOS*), confirming a functional OPDA signaling pathway in liverworts [75]. Similar OPDA biosynthetic genes (*PpLOX6*, *PpAOS1*, *PpAOC1*) have been identified in *P. patens* [21] reflecting the crucial role of OPDA in bryophyte adaptation (Figure 3).

OPDA also mediates growth inhibition in liverworts [75], suggesting a potential adaptive role in stress avoidance of bryophytes (Figure 3). In vascular plants, OPDA induces stress-responsive transcripts, and its ability to enhance heat-shock protein expression in *M. polymorpha* provides strong evidence for its contribution to thermotolerance during early terrestrialization.

MYC, a core TF in JA/OPDA signaling, plays key roles in bryophytes. *Mpmyc* mutants demonstrate that MYC is required for OPDA-mediated growth inhibition, while MYC overexpression leads to pronounced growth suppression, implicating its role in stress adaptation [50]. MYC also activates flavonoid biosynthesis enzyme in *M. polymorpha* (e.g., MpCHALCONE SYNTHASE), an important protective mechanism against environmental stressors.

These findings suggest that bryophytes use MYC-dependent OPDA signaling to mediate abiotic stress responses, including DT. However, the downstream transcriptomic programs remain underexplored, and omics analyses in MYC loss-of-function mutants will be essential to define its precise role. Although ABA–JA interactions are well characterized in angiosperms [76], their relationship in bryophytes is unclear. Investigating ABA-responsive gene expression in OPDA-impaired mutants of *P. patens* and *M. polymorpha* will provide insight into the evolutionary origins of this hormonal crosstalk.

### 3.4. Strigolactones (SLs) as Emerging Regulators in Land Plant Terrestrialization

Strigolactones (SLs) regulate diverse physiological processes—including seed germination, root and shoot growth, nutrient uptake, and stress responses and their signaling is broadly conserved across land plants, from bryophytes to angiosperms [77]. In *P. patens*, the SL biosynthetic genes CAROTENOID CLEAVAGE DIOXYGENASES (CCD7 and CCD8) have been functionally characterized and shown to produce the precursor carlactone (CL) [78]. However, bryophytes lack the angiosperm MORE AXILLARY GROWTH1 (MAX1) ortholog [79], leaving the downstream steps of SL biosynthesis unresolved. The persistence of some SL-like responses in *Ppccd8* mutants suggests the existence of an alternative or ancestral SL biosynthetic pathway in mosses [80].

Canonical SL perception via the receptor DWARF 14 (D14) is absent in bryophytes, whereas KARRIKIN INSENSITIVE2 (KAI2) orthologs are widely conserved in charophytes and bryophytes, indicating that KAI2 may have been the primary receptor in early land plant lineages [79,81]. Although the F-box protein MORE AXILLARY BRANCHES2 (MAX2) is present, the inconsistent phenotypes of *Ppmax2* and *Ppccd8* mutants imply that MAX2 may not operate as a central SL-signaling hub in *P. patens* [82]. Moreover, the function of the putative downstream target SUPPRESSOR OF MAX2-1 LIKE (SMXL) proteins remains unclear in bryophytes. Together, these findings suggest that SL biosynthesis and signaling diverged early between bryophytes and angiosperms. Still, the proposed ancestral KAI2–SMXL module [83] may represent a conserved signaling framework across embryophytes.

SLs commonly inhibit axillary bud outgrowth in angiosperms [84], and a similar inhibitory effect is seen in moss, where SLs reduce caulonemal cell elongation and division [85]. This growth suppression may provide a stress-avoidance strategy in early land plants. In vascular plants, SLs are also involved in drought tolerance [86,87], functioning through both ABA-dependent and ABA-independent pathways [88,89]. Mutant analyses in tomato further support the interdependence of ABA and SL biosynthesis [90,91]. In contrast, the roles of SLs in bryophyte abiotic stress responses, and their potential interaction with ABA signaling, remain unresolved. Investigating SL-deficient and ABA-signaling mutants in bryophytes will be essential to understanding how SLs contributed to early drought tolerance mechanisms.

### 3.5. Karrikins (KARs) as Conserved Regulators of Growth and Stress Responses

Similarly to SLs, karrikins (KARs) influence seed germination, growth, development, and drought tolerance in vascular plants [92]. The core components of KAR signaling—KAI2, MAX2, and SMXL—are conserved in the liverwort *M. polymorpha* [93]. Loss-of-function *Mpkai2a* and *Mpmax2* mutants show reduced thallus growth and altered gemma dormancy, indicating that KAR signaling regulates key developmental transitions in bryophytes [93]. The proposed presence of an endogenous KAI2 ligand (KL) further suggests that both KL–KAI2 and KAR–KAI2 pathways are evolutionarily conserved in bryophytes [93] (Figure 3).

In vascular plants, KAI2 is implicated in drought tolerance by regulating processes such as stomatal closure, cuticle formation, anthocyanin accumulation, and maintenance of membrane integrity [34]. KAR–KAI2-mediated abiotic stress tolerance has been demonstrated in Arabidopsis [94], emphasizing KAI2 as a central regulator. In angiosperms, MAX2 mutants exhibit drought susceptibility due to impaired ABA-mediated stomatal closure [95], supporting a functional interaction between KAR signaling and ABA responses. KARs can also enhance osmolyte accumulation, antioxidant activity, and the expression of ABA-responsive signaling components—including SnRK2.3, SnRK2.6, ABI3, and ABI5—during abiotic stress [96].

However, the roles of KAI2, MAX2, and SMXL in bryophyte stress tolerance remain poorly characterized. The evidence suggests that bryophytes follow KARs signaling to regulate phenotypic alteration reflecting that the mechanisms might be crucial for land plant adaptation [93] (Figure 3). Further transcriptomic analyses of KAI2 and MAX2 mutants exposed to desiccation or ABA will help clarify the contribution of KAR signaling to DT in early land plants and illuminate the evolutionary origins of ABA–KAR crosstalk.

### 3.6. Auxin (AUX) in the Adaptation of Mosses and Liverworts

Auxin (AUX) is a central phytohormone regulating numerous aspects of plant growth, development, and responses to environmental stresses. In vascular plants, increased AUX levels promote seed dormancy under stress, highlighting its role in adaptive responses [97]. However, its evolutionary contribution to early land plant adaptation remains less understood.

Indole-3-acetic acid (IAA), the primary auxin, has been detected in multiple moss species [98]. Core components of AUX biosynthesis, transport, and signaling—including TRYPTOPHAN AMINOTRANSFERASE OF ARABIDOPSIS1 (TAA1), homologues of YUCCA (YUC), AUX transporter gene PIN FOMED (PIN), signaling components genes *TRANSPORT INHIBITOR RESISTANT 1* (*TIR1/AFB*), *Aux/IAA* genes and putative TF encoding gene *AUXIN RESPONSE FACTOR* (*ARF*)—are present in *P. patens* [21], indicating that the auxin regulatory machinery was already established in bryophytes.

As in angiosperms, AUX governs several developmental processes in bryophytes, such as the transition from chloronema to caulonema, rhizoid formation, and cell cycle regulation. In *M. polymorpha*, endogenous AUX also acts as a positive regulator of gemma dormancy [99], consistent with AUX-mediated inhibition of protonemal branching in moss [100]. At the same time, AUX stimulates rhizoid formation [101], suggesting its early adaptive role in promoting anchorage while restricting growth under unfavorable conditions (Figure 3).

To clarify the ancestral contribution of AUX to DT, future studies should examine stress-responsive gene expression in AUX biosynthesis and signaling mutants of key bryophyte models, including *P. patens*, *M. polymorpha*, *S. ruralis*, and *Sphagnum uncinatum*.

### 3.7. Evolutionary Roles of Other Growth Regulators

Several other phytohormones—gibberellins (GAs), cytokinins (CKs), brassinosteroids (Brs), and salicylic acid (SA)—also contribute to stress responses in vascular plants [102,103], but their evolutionary roles in bryophytes remain insufficiently explored.

Although the GA precursor ent-kaurene has been reported in moss [104], bryophytes exhibit non-canonical GA responses, and the function of DELLA proteins is still unclear [21]. Degradation of DELLA protein is critical for GA response but can also be regulated in a GA-independent manner. In angiosperms, DELLA ubiquitination and degradation can be induced by the E3 ubiquitin ligase FLAVIN-BINDING KELCH REPEAT F-BOX1 (AtFKF1) in a GA-dependent manner, and E3 ubiquitin ligase CONSTITUTIVE PHOTOMORPHOGENIC1 (AtCOP1) via ubiquitination in a GA-independent manner [105,106]. Gene co-expression network analysis using orthologs of DELLA-interacting TFs suggests that PpDELLA likely modulates stress responses in bryophytes by repressing plant growth [107]. However, this hypothesis needs to be given further attention to clarify. Although only bryophyte liverworts have *AtSLY1* (encoding F-box protein SLEEPY1) homologs, their function is still to be determined. Bryophytes do not synthesize GAs [108] but presence of orthologs of AtCOP1 in *P. patens* [109], suggesting the GA-independent regulation of DELLA in bryophytes. Further clarification of the interaction of PpCOP1 with PpDELLA might reveal the implication of this vital protein in the early adaptation of land plants. Therefore, further characterizing DELLA function under drought and GA treatments in *P. patens* and *M. polymorpha* might clarify the ancestral GA-related stress mechanisms.

Cytokinins (CKs), known regulators of growth and abiotic stress responses [110], have limited characterization in bryophytes. The cytokinin oxidase gene *PpCKX1*, initially linked to abiotic stress tolerance in Arabidopsis [111], promotes dehydration and salt tolerance in *P. patens* [16]. CKs-driven stimulation of rhizoid development and inhibition of stem growth [15,16] suggests a possible adaptive function in desiccation avoidance (Figure 3).

The presence and functionality of brassinosteroids (Brs) in bryophytes remain uncertain (Figure 3). Although Brs-related responses have been detected in *P. patens*, signaling pathways appear distinct from those in angiosperms [21]. Further transcriptomic analyses are needed to determine whether BRs contributed to early land plant stress resilience.

Similarly, while SA-related responses occur in bryophytes, the pathways underlying SA biosynthesis and signaling remain unclear. Functional studies in model bryophytes are required to understand whether SA played a role in ancestral DT.

## 4. DT in Mosses and Liverworts Is Strongly Supported by the Accumulation of Compatible Solutes

In angiosperms, compounds such as proline, soluble sugars, glycine betaine, and other free amino acids are well-known for helping plants withstand drought and other abiotic stresses [112]. In bryophytes, however, soluble sugars are the most extensively studied osmolytes, and their increased accumulation under stress has been repeatedly documented [113,114].

In the moss *Fontinalis antipyretica*, elevated levels of soluble sugars and compatible inorganic ions contribute to osmoregulation by lowering cellular water potential [115]. The accumulation of proline, soluble sugars and soluble proteins was greatly reported in two moss species, *Hypnum plumaeforme* and *Pogonatum cirratum*, under desiccation stress [116]. Proline and glycine betaine, also reported in mosses, help protect biomolecules by stabilizing hydration shells and maintaining ionic balance [117]. Similarly, *S. uncinata* shows greater accumulation of compatible solutes during desiccation [28]. Mosses such as *Plagiomnium acutum* and *P. patens* also increase levels of altrose, maltitol, ascorbic acid, and proline under drought stress [118], highlighting the evolutionary significance of these solutes in bryophyte stress adaptation (Figure 4).

Abscisic acid (ABA) is a key regulator of osmolyte accumulation in vascular plants [117,119], and a similar mechanism appears in bryophytes. ABA and osmotic stress stimulate soluble sugar accumulation in *P. patens* and *Atrichum androgynum* [22], suggesting that ABA-mediated osmotic adjustment is an ancient strategy for DT. Furthermore, studies in *M. polymorpha* show ABA- and sucrose-induced stress-responsive gene expression and functional characterization of PYL-type ABA receptors [4,20,62], reinforcing ABA’s ancestral role in sugar-related stress signaling.

Collectively, these findings indicate that during cellular desiccation, early land plants either directly accumulate compatible solutes or enhance ABA production, which in turn promotes osmolyte buildup. This protects biomolecules, maintains osmotic balance, and helps preserve membrane integrity (Figure 4).

## 5. Antioxidant Defense in Bryophytes During Cellular Desiccation

Excessive production of reactive oxygen species (ROS)—including singlet oxygen (^1^O_2_), superoxide (O_2_•^−^), hydroxyl radicals (•OH), and peroxides—is a well-documented response to extreme environments in angiosperms [120,121]. Elevated ROS levels disrupt membrane integrity, alter cellular homeostasis, and oxidize macromolecules, ultimately leading to cell death [122]. Similar ROS accumulation has been observed in the mosses *S. uncinata* and *Fontinalis antipyretica* during desiccation [28,115], indicating that oxidative stress is a conserved response across land plants.

To counteract oxidative damage, angiosperms employ a sophisticated antioxidant system composed of non-enzymatic compounds (phenolics, flavonoids, tocopherols, glutathione, ascorbate) and enzymatic antioxidants such as superoxide dismutase (SOD), catalase (CAT), ascorbate peroxidase (APX), and class III peroxidases (POD), which collectively maintain redox homeostasis [54,123]. SOD converts superoxide into H_2_O_2_, which is subsequently detoxified by CAT, APX, and POD.

Although less studied, bryophytes possess similar antioxidant machinery. Exogenous ABA has been greatly demonstrated to induce enzymatic antioxidants like SOD, CAT and POX in *P. patens*, *M. polymorpha* and *A. undulatum* [124]. The induction of SOD, CAT, GST, APX and DHAR by exogenous ABA was also investigated in the gemmalings of *M. polymorpha* [65]. Cytosolic APX has been characterized in the liverwort *Porella lyelli* [35], while SOD was shown to interact with DNA methyltransferase 2 in *P. patens* (PpDNMT2), linking ROS detoxification to epigenetic regulation of stress responses [52]. The role of PpDNMT2 in DT remains a promising research direction. Likewise, the *P. patens* gene encoding monodehydroascorbate reductase (MDHAR) supports a role for redox enzymes in stress adaptation [33]. Reduced expression of CAT and dehydroascorbate reductase (DHAR) in *M. polymorpha* TCP-P mutants (*Mptcp1ge*) highlights a potential early regulatory function of TCP in bryophyte antioxidant control [49]. Since, TCP-P and TCP-C are essential for cell proliferation and flavonoid biosynthesis, respectively, the presence of both major TCP clades in *Marchantia* [3] further suggests an evolutionary foundation for this regulatory network (Figure 4).

Enzymatic antioxidant activity—including SOD, CAT, APX, glutathione S-transferase (GST), DHAR, and POD—rises significantly in *S. uncinate* and *M. polymorpha* during desiccation [28,65], underscoring their importance in bryophyte DT and early terrestrial adaptation.

In addition to enzymatic defenses, bryophytes accumulate non-enzymatic antioxidants including diverse secondary metabolites that bolster oxidative protection (Figure 4). Flavonoid biosynthesis is widely induced under stress [125,126]. The UVR8-mediated pathway has been described in *P. patens* and *M. polymorpha* [44,45]. Moreover, the abiotic-stress-responsive TF bHLH1 enhances flavonoid production in *P. appendiculatum* [51], highlighting a potential role for the JA/OPDA–bHLH1 axis in bryophyte stress adaptation—although its connection to ABA remains unknown. Mosses also accumulate high carotenoid levels [127] and β-carotene ketolase genes (*CrBKT* and *HpBHY*) in *P. patens* are upregulated under heat, oxidative stress, and ABA treatment [48], illustrating an ancient photoprotective and antioxidant function.

The involvement of MAPKKKs, particularly Raf B3, which strongly enhance antioxidant activity in angiosperms [128] and participate in ABA signaling across land plants [5], remains unexplored in bryophytes and represents a significant knowledge gap (Figure 4).

Together, these findings demonstrate that DT in bryophytes relies heavily on the activation of antioxidant defenses—both enzymatic and metabolic—directly triggered by water loss or mediated by ABA signaling. These systems protect membranes, limit lipid peroxidation, and maintain cellular redox balance. The conserved ROS-detoxification pathways observed in bryophytes likely represent an evolutionary framework that enabled early land plants to survive extreme terrestrial environments (Figure 4). Future omics-driven studies focusing on ROS signaling in models such as *P. patens* and *M. polymorpha* will further clarify the evolutionary origins of antioxidant machinery in land plant adaptation

## 6. Conclusions and Future Perspectives

The evolution and adaptation of bryophytes involve a highly coordinated regulatory network shaped by diverse morphological, physiological, biochemical, and molecular processes. Their widespread constitutive vegetative DT stems from the action of stress-responsive components—including phytohormones, protective proteins, osmolytes, and antioxidants—that collectively support survival in extreme terrestrial environments. Following cellular dehydration induced by drought, salinity, osmotic stress, or temperature extremes, bryophytes rapidly accumulate ABA. This activates an ABA-dependent signaling cascade involving receptors, upstream kinases, and downstream TFs, ultimately enhancing the expression of ABA- and drought-responsive genes that safeguard essential cellular structures.

Cellular desiccation and ABA also promote the accumulation of compatible solutes—such as soluble sugars, proline, and glycine betaine—which facilitate osmotic adjustment, stabilize membranes, and aid in rapid rehydration. Concurrently, bryophytes enhance antioxidant defenses to efficiently detoxify reactive oxygen species generated during dehydration. In addition to ABA-dependent pathways, ABA-independent mechanisms, including DREB-mediated gene expression, further strengthen tolerance to water-limiting conditions. Other phytohormones, such as ET, JA, SL, KAR, AUX, and CK also modulate growth suppression and contribute to desiccation avoidance, although their roles in bryophyte DT remain less understood.

While ABA signaling has been relatively well characterized in mosses and liverworts, the desiccation-responsive functions of other hormones require deeper investigation. Future studies using biosynthetic and signaling mutants in model bryophytes (*P. patens* and *M. polymorpha*), combined with multi-omics approaches, will help uncover additional DT mechanisms. Moreover, examining fully desiccation-tolerant species such as *S. ruralis* and *S. uncinata* holds promise for revealing the complex evolutionary strategies that enabled bryophytes to thrive in harsh terrestrial environments.

## Figures and Tables

**Figure 2 ijms-27-00478-f002:**
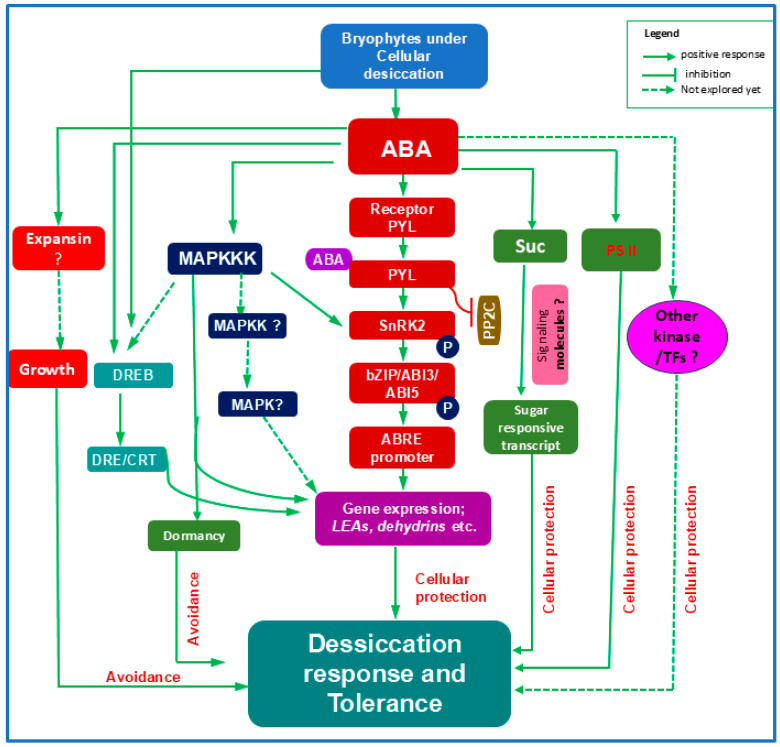
The phytohormone ABA plays a central role in enabling desiccation tolerance in bryophytes through a complex regulatory network. Cellular dehydration increases endogenous ABA levels, which activates the canonical ABA signaling cascade involving PYL receptors, PP2C phosphatases, SnRK2 kinases, bZIP transcription factors, and ABRE promoter elements to induce stress-responsive gene expression. ABA also stimulates B3-type Raf MAPKKKs, which further enhance SnRK2 activity, although the contributions of downstream MAPKK and MAPK modules to bryophyte desiccation tolerance remain unresolved. ABA promotes the accumulation of sucrose and sugar-responsive transcripts, though the specific sugar-signaling components in bryophytes are still unknown. Additionally, the DREB transcription factor, activated by desiccation or ABA, binds DRE/CRT motifs to drive desiccation-related gene expression. ABA contributes to photoprotection by improving PSII recovery and suppresses growth—part of the desiccation avoidance strategy—potentially through regulation of growth-related proteins such as expansins, whose mechanistic roles are not yet clarified. Other ABA-responsive transcription factors may also contribute to tolerance, though their functions remain to be explored. Abbreviations: ABA, abscisic acid; ABI3, abscisic acid insensitive 3; ABRE, ABA-responsive element; bZIP, basic leucine zipper; CRT, C-repeat element; DRE, dehydration-responsive element; DREB, DRE-binding protein; LEA, late embryogenesis abundant; MAPKKK, mitogen-activated protein kinase kinase kinase; PP2C, protein phosphatase 2C; PYL, pyrabactin like; SnRK2, sucrose non-fermenting 1-related protein kinase; Suc, sucrose; PSII, photosystem II; TFs, transcription factors.

**Figure 3 ijms-27-00478-f003:**
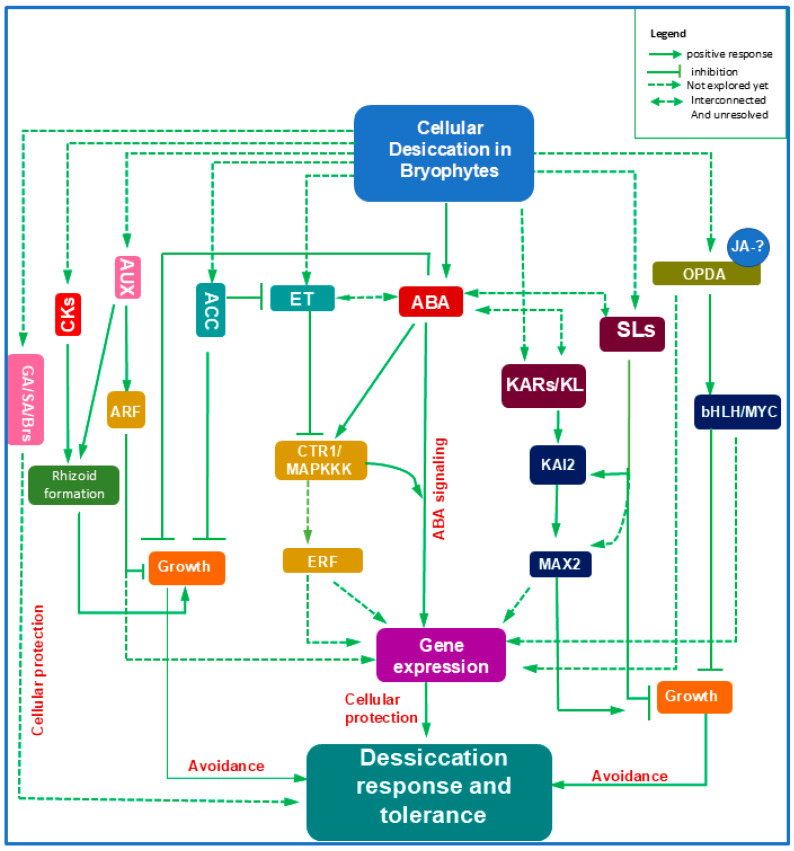
Crosstalk among multiple phytohormones contributes to desiccation tolerance in bryophytes. Abscisic acid (ABA) serves as the central regulator, with desiccation-induced ABA accumulation coordinating downstream pathways and interacting with other hormones to enhance stress tolerance. Ethylene (ET) and its precursor ACC may operate through a separate growth-inhibitory route, where CTR1/MAPKKK-mediated signaling modulates potentially interfaces with ABA responses, although the exact ABA–ET interaction in bryophytes remains unclear. The jasmonic acid (JA) precursor OPDA may promote desiccation tolerance by activating stress-responsive genes, while JA-associated TFs such as MYC/bHLH could further enhance stress-related transcription, although their roles are not yet fully resolved. Strigolactones (SLs) and karrikins (KARs/KL) may participate in KAI2–MAX2-dependent modulation of growth; however, their contribution to stress-responsive gene expression remains largely unexplored. These pathways may still connect with ABA signaling, suggesting coordinated regulation during stress acclimation. Bryophytes also use AUX–ARF signaling to suppress growth, while AUX and cytokinins (CKs) jointly regulate rhizoid formation, supporting early terrestrial adaptation. In contrast, the stress-related functions of gibberellins (GAs), brassinosteroids (BRs), and salicylic acid (SA) in bryophytes remain poorly understood. Altogether, desiccation tolerance in bryophytes likely arises from integrated cellular protection and avoidance mechanisms mediated through the interplay of diverse phytohormones networks. Abbreviations: ACC, aminocyclopropane-1-carboxylate; ARF, auxin response factor; AUX, auxin; bHLH, basic helix-loop-helix; Brs, brassinosteroids; CKs, cytokinins; CTR1, constitutive triple response 1; GAs, gibberellic acids; ERF, ethylene-responsive factor; ET, ethylene; JA, jasmonic acid; KARs, karrikins; KAI, karrikin insensitive; KL, karrikin-insensitive ligand; MAPKKK, mitogen-activated protein kinase kinase kinase; MAX, more axillary growth; OPDA, 12-oxo-10,15(Z)-phytodienoic acid; SA, salicylic acid; SLs, strigolactones.

**Figure 4 ijms-27-00478-f004:**
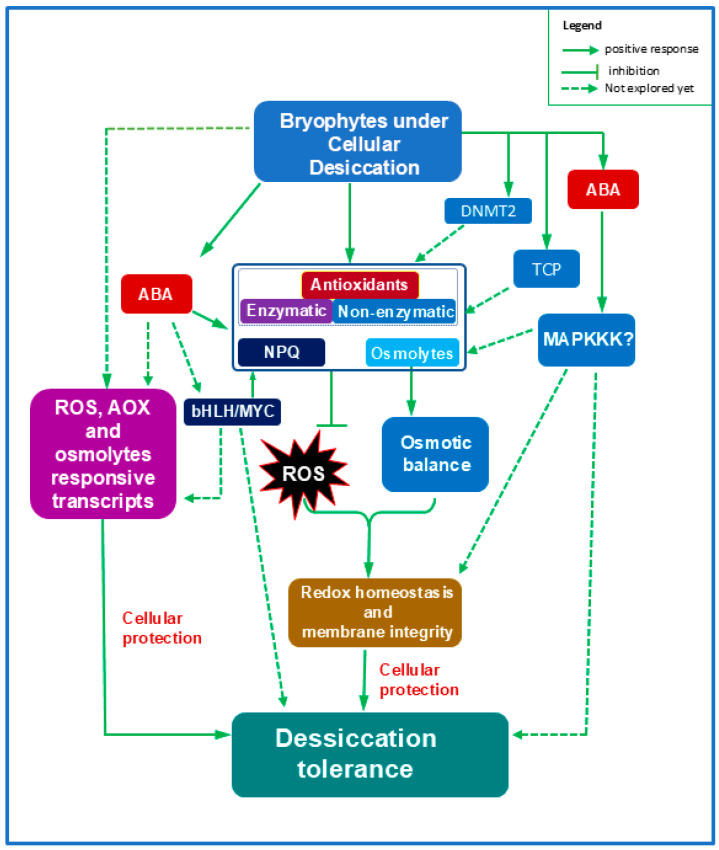
Desiccation tolerance in bryophytes is supported by enhanced osmotic adjustment and redox balance. Desiccation stress or ABA activates a broad antioxidant network—including enzymatic and non-enzymatic antioxidants, NPQ, and osmolytes—that detoxifies ROS, protects cellular components, and maintains osmotic stability, ultimately preserving redox homeostasis and membrane integrity. ABA may also recruit stress-responsive transcription factors such as bHLH/MYC to stimulate antioxidant defenses and inhibit growth. Whether bryophytes utilize B3 Rafs (MAPKKKs) to reinforce antioxidant pathways, as documented in angiosperms, remains unclear. Overall, ROS-mediated antioxidant systems appear to be conserved in bryophytes during cellular desiccation. Abbreviations: ABA, abscisic acid; MAPKKK, mitogen-activated protein kinase kinase kinase; NPQ, non-photochemical quenching; bHLH, basic helix-loop-helix; TCP, TEOSINTE BRANCHED1–CYCLOIDEA–PROLIFERATING CELL FACTOR; DNMT2, DNA methyltransferase 2.

**Table 1 ijms-27-00478-t001:** Characterized signaling components in bryophytes mosses and liverworts in response to desiccation stress.

Signaling Components	Bryophyte Species	Characteristics	Stress Responses	References
PpMDHAR1, PpMDHAR3	*P. patens*	Monodehydro ascorbate reductase, enzyme for Ascorbate-glutathione cycle for regulating ROS	Drought, osmotic and salinity stress	[33]
PpDHNA, PpDHNC	*P. patens*	Dehydrin, Group 2 LEA having role to abiotic stress adaptation	ABA, drought and osmotic stress	[34]
PpDBF1	*P. patens*	Dehydration responsive elements (DRE)binding transcription factor	ABA, drought and salt stress	[34]
APX	*Pallavicinia lyelli*	Ascorbate peroxidase, enzymatic antioxidant for scavenging ROS	oxidative stress	[35]
LEA	*P. patens* and *M. polymorpha*	Late embryogenic abundant proteins	Desiccation stress, osmotic stress and ABA	[20]
PpABI3	*P. patens*	Abscisic acid insensitive 3, Transcription factor interacting to ABI5 and showing ABA signaling	ABA, desiccation, cold and oxidative stress	[36]
PpABA1/MpABA1	*P. patens* and *M. polymorpha*	Zeaxanthin Epoxidase (ZEP), ABA biosynthesis gene	ABA and hyperosmotic responses	[22,37]
NCED	*P. patens*	9-cis-epoxycarotenoid dioxygenase, ABA biosynthesis gene	ABA, dehydration and osmotic stress	[38]
AAO	*P. patens*	Aldehyde oxidase, ABA biosynthesis gene	ABA and dehydration	[38]
ABI5	*P. patens*	Abscisic acid insensitive 5, bZIP like transcription factor having wide array of abiotic stress responses	ABA, dehydration and osmotic stress	[38]
PpARK/ANR/ PpCTR1/MpARK	*P. patens* and *M. polymorpha*	Raf-like kinase (Mitogen activated protein kinase kinase kinase), ABA and ET signaling components	ABA, dehydration and hyperosmotic stress	[5,38,39,40,41]
ACGT core motif	*M. polymorpha*	Cis-acting promoter elements	ABA	[20]
PpABI1, MpABI1A and MpABI1B	*P. patens* and *M. polymorpha*	Abscisic acid insensitive protein phosphatase 2 C, Negative regulator of ABA signaling	ABA	[42,43]
UVR8	*P. patens* and *M. polymorpha*	UV resistance locus 8 photoreceptor, Flavonoid biosynthesis gene, non-enzymatic antioxidant	UV B stress/light stress, oxidative stress	[44,45]
PpSnRK2	*P. patens*	Sucrose non-fermenting 1 related protein kinase, regulating wide array of stress responses	ABA and desiccation stress	[46]
PYR1, PYL	*P. patens* and *M. polymorpha*	Pyrabactin resistance 1, Pyrabactin like receptor for ABA signaling	ABA and osmotic stress	[4,47]
CrBKT	*P. patens*	β-carotene ketolase for carotenoid biosynthesis	Heat stress, oxidative stress and ABA	[48]
MpTCP1	*M. polymorpha*	Teosinte branched1/Cincinnata/proliferating cell factor (TCP), bHLH TF	Cell proliferation, Oxidative stress, Redox homeostasis	[49]
MpMYC	*M. polymorpha*	bHLH TF, Jasmonic acid pathway	JA signaling	[50]
PabHLH	*M. polymorpha*	Basic helix-loop-helix, Flavonoid synthesis, bibenzyls synthesis	UV stress and SA	[51]
PpDNMT2	*P. patens*	DNA methyltransferase2	Salinity and oxidative stress responses	[52]
MpCOI1	*P. patens*	Coronatine insensitive1, JA signaling component	JA, OPDA and heat stress	[53]
PpCKX1	*P. patens*	Cytokinin oxidase/dehydrogenase, enzyme for cytokinin degradation	Dehydration and salinity stress	[16]

Abbreviations: AAO, aldehyde oxidase; ABI, abscisic acid insensitive; APX, ascorbate peroxidase; BKT, β-carotene ketolase bHLH, basic helix-loop-helix; CKX1, cytokinin oxidase 1; COI, coronatine insensitive; DBF, dehydration responsive element binding factor; DHN, dehydrin; DNMT, DNA methyltransferase; LEA, late embryogenic abundant proteins; MDHAR, monodehydro ascorbate reductase, NCED, 9-cis-epoxycarotenoid dioxygenase; PYL, pyrabactin like; PYR, pyrabactin resistance; SnRK2, sucrose non-fermenting 1 related protein kinase; TCP, teosinte branched1cincinnata proliferating; UVR, UV resistance.

## Data Availability

No new data were created or analyzed in this study. Data sharing is not applicable to this article.

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
