# Peer review of "Desiccation Tolerance in Moss and Liverwort: Insights into the Evolutionary Mechanisms of Terrestrialization"

_ijms, 2026, doi:10.3390/ijms27010478_

Round 1

Reviewer 1 Report

Comments and Suggestions for Authors

In this piece of review, the authors critically reviewed the mechanistic insights of bryophytes particularly mosses and liverworts during their terrestrial adaptation which have great values for better understanding of land plant adaptation in extreme habitats. They discussed on the phytohormones, osmotic balance and redox homeostasis of bryophytes under cellular desiccated condition induced by different abiotic stresses like drought and osmotic stresses. Their discussion and analyses highlighted the conserved molecular mechanisms of bryophytes under desiccation stress, what are really appreciating. However, further improvements are needed in some points to make strong grounds of the authors’ claims.

  1. In table 1, the authors mostly emphasized on two model bryophytes P. patens and M. polymorpha. Addition of data of others species of mosses and liverworts should improve their discussion and analysis ground in terms of terrestrialization of bryophytes.
  2. The mechanisms of ABA-mediated growth arrest in P. patens are well clarified. The authors should discuss more in this respect to clarify the mechanistic roles of ABA in inhibiting growth during terrestrialization of land plants.
  3. The authors did not critically analysed ABA and ET crosstalk in bryophytes in terms of desiccation tolerance, which need to be updated further.
  4. While discussing with GA response, the authors claimed that DELLA protein has transcriptional roles in P. patens. What transcriptional roles likely to be modulated by DELLA in bryophytic adaptation was not manifested in this part.
  5. The authors should check the scientific name throughout the manuscripts so that they don’t use the genus name repeatedly rather than abbreviated form. Scientific names were not followed by italics in some cases also.
  6. Although ABA has crucial roles in modulating antioxidant enzymes in angiosperms, such evidences were not critically discussed while the authors explaining the oxidative stress in bryophytes.
  7. Authors should be very careful to use abbreviated form of word after first uses. Sometimes they used abbreviated form and sometimes full terms.
  8. The authors should improve their figures quality particularly fig 2 and 3 which are little bit complex and clumsy and not self-explanatory. The authors could omit the incidence of AUX, CKs, GA and others from the figure 3 which have very negligeable evidences regarding desiccation tolerance in bryophytes. This is not mandatory that all information must be included in the figures.

Reviewer 2 Report

Comments and Suggestions for Authors

This review systematically elucidates the survival strategies of bryophytes as early terrestrial representatives, demonstrating how they integrate multiple molecular mechanisms to adapt to drought stress. The study not only reveals the evolutionary significance of their unique desiccation tolerance, but also provides a critical theoretical foundation for understanding environmental adaptation mechanisms during plant terrestrialization. However, this manuscript still needs some modification before publication.

Minor:

Line 61 Figure 1

In my view, while Figure 1 aims to illustrate how root development facilitated adaptation to dehydration stress throughout the evolutionary trajectory from bryophytes to angiosperms, this theme appears weakly connected to the article's central thesis—specifically, how mosses and liverworts leverage phytohormones, stress-responsive proteins, compatible solutes, antioxidants, and integrated signaling networks to survive in arid terrestrial environments. Rather than clarifying, this approach may misleadingly associate vegetative desiccation tolerance with root evolution. If the goal is to emphasize the importance of rhizoids in bryophyte stress adaptation, I recommend focusing the figure exclusively on bryophyte rhizoid systems.

Line 99

"Desiccation tolerance" should not be italicized.

Line 115

Repeat headers on continued tables.

All Figures

The resolution of the images in the article is somewhat low. If possible, please replace them with higher-resolution images.

Line 241

The MpAOC and MpAOS should be italicized. Gene names should be italicized; please also verify this in other sections of the article.

Line 315 “Current evidence suggests that bryophytes may utilize KARs to reg- 315 ulate growth suppression and dormancy—traits crucial for desiccation avoidance”

.Provide citations for "recent evidence" in this sentence.
